# *OsGSTU17*, a Tau Class Glutathione S-Transferase Gene, Positively Regulates Drought Stress Tolerance in *Oryza sativa*

**DOI:** 10.3390/plants12173166

**Published:** 2023-09-04

**Authors:** Jinyan Li, Lijun Meng, Shuohan Ren, Chunying Jia, Ruifang Liu, Hongzhen Jiang, Jingguang Chen

**Affiliations:** 1School of Agriculture, Shenzhen Campus of Sun Yat-sen University, Shenzhen 518107, China; lijy697@mail2.sysu.edu.cn (J.L.); renshh@mail3.sysu.edu.cn (S.R.); pureying@126.com (C.J.); 2Kunpeng Institute of Modern Agriculture at Foshan, Foshan 528225, China; menglijun@caas.cn; 3The High School Affiliated to Renmin University of China, Shenzhen 518119, China; liuruifang@rdfzsz.cn

**Keywords:** *OsGSTU17*, drought tolerance, glutathione S-transferases, CRISPR/Cas9, *Oryza sativa* L.

## Abstract

As a great threat to the normal growth of rice, drought not only restricts the growth of rice, but also affects its yield. Glutathione S-transferases (GSTs) have antioxidant and detoxification functions. In rice, GSTs can not only effectively cope with biological stress, but also play a defense role against abiotic stress. In this study, we selected *OsGSTU17*, a member gene that was induced by drought, to explore the role of GSTs and analyze their physiological mechanisms that are involved in rice drought tolerance. With the CRISPR/Cas9 knockout system techniques, we obtained two independent mutant lines of *osgstu17*. After 14 days of drought stress treatment, and then re-supply of the water for 10 days, the survival rate of the *osgstu17* mutant lines was significantly reduced compared to the wild-type (WT). Similarly, with the 10% (*w*/*v*) PEG6000 hydroponics experiment at the seedling stage, we also found that compared with the WT, the shoot and root biomass of *osgstu17* mutant lines decreased significantly. In addition, both the content of the MDA and H_2_O_2_, which are toxic to plants, increased in the *osgtu17* mutant lines. On the other hand, chlorophyll and proline decreased by about 20%. The activity of catalase and superoxide dismutase, which react with peroxides, also decreased by about 20%. Under drought conditions, compared with the WT, the expressions of the drought stress-related genes *OsNAC10*, *OsDREB2A*, *OsAP37*, *OsP5CS1*, *OsRAB16C*, *OsPOX1*, *OsCATA*, and *OsCATB* in the *osgtu17* mutant lines were significantly decreased. Finally, we concluded that knocking out *OsGSTU17* significantly reduced the drought tolerance of rice; *OsGSTU17* could be used as a candidate gene for rice drought-tolerant cultivation. However, the molecular mechanism of *OsGSTU17* involved in rice drought resistance needs to be further studied.

## 1. Introduction

Global agricultural production is severely restricted by drought stress; this constraint is more obvious in warm arid and semi-arid regions [1]. By 2080, crop yield reduction caused by drought stress is expected to directly or indirectly affect the living environment of most of the world’s population [2]. In recent years, possibly due to the impact of global warming, droughts are occurring more frequently and with greater severity [3].

Drought, or water shortage in plants, can affect many physiological processes and parameters by altering the osmotic pressure within the plant. This can exert influence over both the enzymatic and non-enzymatic physiological processes, including stomatal conductance, membrane electron transport chains, photosynthetic efficiency reduction, and modulation of reactive oxygen species (ROS) production and scavenging mechanisms [4]. Among them, the ROS include superoxide free radical (O_2_^−^), singlet oxygen (^1^O_2_), hydrogen peroxide (H_2_O_2_), and hydroxyl free radical (OH^.^). Under normal physiological conditions, the level of ROS produced in each organelle is normal, but under environmental stress conditions, the production of ROS increases several times [5,6]. Different levels of H_2_O_2_ have different effects on plants. When the accumulation of H_2_O_2_ in plant cells is low, the defensive signaling pathways in plants are activated. At high levels, H_2_O_2_ is harmful to cells and leads to programmed cell death [5,7]. Excess ROS also leads to impaired membrane lipid and malondialdehyde (MDA) production, ultimately leading to protein degradation, disruption of the membrane fluidity, and inhibition of ion transport. MDA is generally regarded as a key indicator of the plant oxidative damage level; the level of the plant oxidative damage simultaneously increases when the MDA content increases [8].

Plants have evolved a mechanism to remove excess ROS through enzymatic or non-enzymatic antioxidants. The enzymes in plants included catalase (CAT), glutathione peroxidase (GPX), guaiacol peroxidase (POX), glutathione S-transferase (GST), ascorbate peroxidase (APX), and so on. Non-enzymatic antioxidants include ascorbic acid (AsA), glutathione (GSH), carotenoids (Car), phenolic compounds (total content of polyphenols: TPC), and proline (Pro) [6,8,9]. Among the non-enzymatic antioxidants is a tripeptide, glutathione (GSH) (γ-Glu-Cys-Gly), which acts as an antioxidant to make plant cells resistant to ROS-induced oxidative damage [6]. The expression of *OsGSTU17* showed significant changes under different abiotic stress treatments, proving that *OsGSTU17* has a variety of roles in plant, including helping plants resist external environmental stress, regulation of the cellular redox balance, regulation of stress-related gene expression, exogenous detoxification, sulfhydryl group protection, and influencing the enzyme activity [10]. At the same time, it can also mediate protein and nucleotide biosynthesis, engage in the sequestration of heavy metals, and contribute to the regulation of plant growth and senescence [6].

The multiple effects of glutathione in plants suggest that glutathione levels and metabolism may be related to plant stress resistance. The role of glutathione in plants is often regulated by the catalysis of glutathione S-transferases (GSTs). In plants, the role of GSTs is similar to that of glutathione peroxidase, as GSTs can reduce the production of peroxides during the oxidation process [11]. GSTs can be divided into six families, among which tau (GSTU) and phi (GSTF) class GSTs are plant specific and are the most representative of the six families [12]. We found that the glutathione S-transferases gene *OsGSTU17* was induced by drought stress, so we further investigated the role of *OsGSTU17* in the drought tolerance of rice.

## 2. Results

### 2.1. Relative Expression of OsGSTU17 under Drought Stress

To investigate whether *OsGSTU17* is involved in the drought tolerance of rice, we simulated drought by using 10% PEG6000 (Macklin, Shanghai, China) solution and analyzed whether its expression was induced. The results showed that compared with the before treatment, the application of 10% PEG6000 significantly induced a notable upregulation in the expression of *OsGSTU17* in both the aerial shoot and the root tissues of the plants (Figure 1). In the root, after the 10% PEG6000 treatment, the expression of *OsGSTU17* increased significantly with increased time, and after 12 h, the expression reached the highest level, approximately 5.3 times higher than before the treatment (Figure 1A). In the above-ground part, *OsGSTU17* expression reached its highest level at 24 h after treatment, about 4.8 times higher than before the treatment (Figure 1B). These results suggest that the *OsGSTU17* gene is likely involved in rice drought tolerance.

### 2.2. Study on Drought Sensitivity of OsGSTU17 Transgenic Line at Seedling Stage

To further verify whether *OsGSTU17* affects the drought resistance of rice, we obtained the mutant lines *osgstu17-1* and *osgstu17-2* using CRISPR/Cas9 technology (Figure 2A). Drought stress tests were carried out on the mutant strain and wild type (WT) at seedling stage. We first provided water for 21 days for the normal growth of rice, and then stopped the water supply; we re-watered after 14 days of discontinuation, and calculated the survival rate 10 days later (Figure 2B). We calculated the survival rate by using the criterion of complete leaf regrowth as the standard for survival; the survival rate of the WT was 14.2%, while that of *osgstu17-1* and *osgstu17-2* were 2.1% and 2.0%, which were significantly lower than that of the WT (Figure 2C). These results indicated that the drought tolerance of rice was significantly reduced after the deletion of the *OsGSTU17* gene.

### 2.3. Effects of OsGSTU17 Knockout on Rice Seedling Growth

To further evaluate the effect of the *OsGSTU17* gene on rice drought stress, we grew the transgenic *osgstu17-1* and *osgstu17-2* strains seedlings in normal IRRI solution for 14 days, subsequently transferred into nutrient solution containing 10% PEG6000 for 14 days, set up a control by using nutrient solution without PEG6000, and measured the dry weight of the mutant and WT (Figure 3A,B).

### 2.4. Effects of *OsGSTU17* Knockout on Rice Seedling Physiological Indicators

After analyzing the dry weight results, we found that the biomass of the *osgstu17* mutant strain in normal solution was not significantly different from that of the WT. However, under drought stress, the average root dry weight of the *osgstu17* mutant strains decreased by 43.4% compared to the WT (Figure 3C), and the shoot dry weight decreased by 39.3% (Figure 3D).

We also measured some physiological indicators associated with drought stress. In comparison to the wild type (WT), no noteworthy disparities were observed in terms of chlorophyll content, proline content, MDA and H_2_O_2_ content, and CAT and SOD activity of the *osgstu17* mutant strains (Figure 4). But under 10% PEG6000 treatment, compared with the WT, the chlorophyll content of the *osgstu17* mutant was decreased by 24.1% on average (Figure 4A), the proline content was decreased by 22.7% (Figure 4B), and the average MDA content was increased by 23.7% (Figure 4C). On average, the H_2_O_2_ content increased by 20.9% (Figure 4D), the CAT activity decreased by 17.1% (Figure 4E), and the SOD activity decreased by 19.3% (Figure 4F).

### 2.5. Effects of OsGSTU17 Knockout on Expression of Related Stress Genes

We conducted an analysis of the expression patterns of several well-established genes implicated in both abiotic and biotic stress responses within the transgenic lines of *OsGSTU17* to explore the role of *OsGSTU17* in drought stress response; among these genes were *OsNAC10*, *OsDREB2A*, *OsAP37*, *OsP5CS1*, *OsRAB16C*, *OsPOX1*, *OsCATA*, and *OsCATB* [13,14,15,16]. The result showed that, compared with the WT, there was no significant difference in the above-ground expression of all these genes when under normal conditions (Figure 5A), but under drought stress, the expression of all these genes in the above-ground of the *OsGSTU17* transgenic line was significantly decreased compared with the WT (Figure 5B).

## 3. Discussion

Rice (*Oryza sativa* L.) is one of the most important food crops, with more than half of the world’s population consuming rice as a staple food. The consumption and production of rice hold a significant proportion in the entire food supply chain, directly affecting people’s survival and safety. Therefore, it is necessary to increase rice production to meet the growing needs of the population, as described in the following discussion. Although rice has been widely considered as the staple food of most people, the harm of drought to rice has not been completely solved, and a certain section of rice-growing area has been threatened by drought. Previous studies have shown that plant growth and nitrogen metabolism are strongly affected by drought stress. Drought affects the osmotic imbalance of rice, the enzymatic and non-enzymatic biochemical pathways, stomatal conductance, membrane electron transport chains, photosynthesis reduction, and impacts the generation and detoxification of reactive oxygen species (ROS) [14,15]. Glutathione S-transferases (GSTs) are key metabolic enzymes in rice that catalyze the binding of glutathione to various electrophilic compounds, which can effectively inhibit the damage of drought to rice [11].

Pavlidi et al. [17] reported that GSTs play a crucial role in detoxification and antioxidant of hydrophobic substances in and out of plants. Srivastava et al. [18] also reported the role of GSTs in resistance to heavy metal stress. Chan and Lan [19] reported that the expression of *GmGSTL1* in soybeans is up-regulated under salt stress, and this ability to protect plants under salt stress may be due to its interaction between the antioxidant flavonoids quercetin and kaempferol. In rice, Kim et al. [20] reported that the presence of a naturally occurring allelic variant of *OsGSTZ2* was associated with reduced cold hardiness. Kumar et al. [21] reported that the expression of *OsGSTL2* effectively inhibited arsenate stress, drought stress, and cold stress. *OsGSTU12* is involved in regulating the senescence of rice leaves, and the overexpression of *OsGSTU12* delays the senescence of rice leaves [22]. Jing et al. [23] showed that *OsGSTU6* was involved with the intracellular ROS homeostasis in rice; it may play an important role in Cd stress tolerance. *OsGSTU5* confers tolerance against arsenic toxicity in rice by accumulating more arsenic in the root [24]. Overexpression of *OsGSTU30* also can enhance the drought stress tolerance in *Arabidopsis* [18]. qRT-PCR analysis revealed that the expression level of *OsGSTU17* under drought stress was significantly induced by drought stress (Figure 2), indicating its potential involvement in drought tolerance in rice. Furthermore, through pot drought experiments, we found that the survival rate of the *OsGSTU17* knockout mutant after the drought significantly decreased compared with the WT (Figure 3B), indicating that *OsGSTU17* is involved in the drought tolerance of rice.

Under drought stress treatment, compared with the WT, the shoot biomass of the *osgstu17* mutant lines decreased by 39.3%, the MDA content and H_2_O_2_ content of the transgenic lines increased by 23.7% and 20.9% (Figure 4C,D), and the proline content, CAT activity and SOD activity of *osgstu17* mutant lines decreased by 22.7%, 17.1%, and 19.3% (Figure 4B,E,F). Proline serves as a crucial modulator of plant resilience in the face of diverse stressors, including drought and elevated salinity [25]. MDA content is an important index of cell damage in plant stress [26]. CAT and SOD can decompose H_2_O_2_, regulate reactive oxygen species (ROS) homeostasis, and thereby improve plant resistance to stress [27]. Under drought stress, the accumulation of H_2_O_2_ in rice increased; the accumulation of H_2_O_2_ can lead to cell damage and eventually cell death [28]. One function of GSTs is to remove ROS, including superoxide radicals, singlet oxygen, alkoxy radicals, hydroxyl radicals, and hydrogen peroxide [10,11,15,22]. The above results indicate that *OsGSTU17* may improve the adaptability of rice to drought stress by clearing ROS.

Finally, this experiment completed the detection of the expression levels of several typical drought-resistant genes under drought conditions. Among them, Jeong et al. [12] found that the overexpression of *OsNAC10* in rice can enhance the drought tolerance, high salinity, and low temperature in the vegetative stage, and significantly improve the drought resistance during the breeding period. *OsDREB2A*, a member of the DREBP subfamily of rice AP2/ERF transcription factors, may be involved in abiotic stress responses by directly binding to DRE elements to regulate downstream gene expression, thus exhibiting drought resistance [13]. *OsAP37* is a phosphorylation target of growth under drought kinase, which also regulates rice growth under drought [14]. In addition, *OsP5CS1* is related to the osmotic regulation of rice and can regulate proline expression [15]. After knocking out *OsGSTU17*, the expression levels of the above genes were down-regulated.

In conclusion, through pot and hydroponics experiments on the mutant lines of *osgstu17*, we found that *OsGSTU17* is involved in rice drought tolerance. This gene can be considered as a candidate gene for rice drought tolerance breeding, but its molecular mechanism still needs further exploration.

## 4. Materials and Methods

### 4.1. Plasmid Construction and Plant Transformation

The sgRNA-Cas9 vector was meticulously generated by synthesizing oligonucleotides that corresponded to the guide RNA sequences specifically targeting *OsGSTU17*. These synthesized oligos were subsequently integrated into the CRISPR/Cas9 genome editing vector, resulting in the establishment of the functional sgRNA-Cas9 construct. Primer design: 5′-ggcaGAGCCCCATGGCGATCCGTG-3′ and 5′-aaac CACGGATCGCCATGGGGCTC-3′.

Nipponbare rice cultivar was employed as the host in agrobacterium-mediated transformation, aiming to generate CRISPR-induced mutant lines targeting the *osgstu17* locus. For the purpose of identifying mutations within the generated mutant lines, total DNA was extracted from transgenic plant material and subsequently utilized as a template for amplification of the genomic regions encompassing the CRISPR target sites, employing primers designed for this purpose. Subsequently, the resulting PCR amplicons were subjected to Sanger sequencing for mutation detection. Validation of all plasmid constructs was undertaken through sequencing analysis conducted by Sangon Biotech, Shanghai, China.

### 4.2. Plant Materials and Growth Conditions

To assess the seedling survival under drought stress, a batch of seedlings was cultivated for a duration of 21 days under adequately watered conditions, ensuring the presence of 10 seedlings of uniform size in each pot. Subsequently, irrigation was suspended for a period of 14 days, followed by a 10-day rewatering phase.

For the hydroponic drought stress investigations, the rice seedlings were initially cultivated using the standard IRRI solution for a span of 2 weeks. Thereafter, the seedlings were transplanted into a nutrient solution supplemented with 15% (*w*/*v*) PEG6000 for an additional 2 weeks to induce drought stress. These hydroponic experiments were conducted within a controlled growth environment, maintaining a photoperiod of 14 h of light (30 °C) from 8:00 to 22:00, followed by 10 h of darkness (22 °C) from 22:00 to 8:00, accompanied by a relative humidity of 60%. In all the experimental conditions, the nutrient solutions were renewed at 48 h intervals.

### 4.3. Genes Expression Analysis

Total RNA was isolated utilizing the TRIzol reagent (Vazyme Biotech Co., Nanjing, China). Subsequently, DNase I-treated total RNAs underwent reverse transcription (RT) employing the HiScript II Q Select RT SuperMix for qPCR (+gDNA wiper) kit (Vazyme Biotech Co.). Replicate quantitative analyses were conducted using the 2 × T5 Fast qPCR Mix (SYBRGreenI) kit (TsingKe Co., Beijing, China). The primers employed for the quantitative real-time polymerase chain reaction (qRT-PCR) are provided in Table 1.

### 4.4. Chlorophyll Content

The extent of relative chlorophyll content in the most recently fully expanded leaf was assessed employing an SPAD-502 Chlorophyll Meter (Minolta Co., Tokyo, Japan).

### 4.5. Proline Content, MDA Content, H_2_O_2_ Content, CAT Activity and SOD Activity

The proline content was determined according to Kavi Kishor and Sreenivasulu [18]. Malondialdehyde (MDA) content was quantified using the procedure outlined by the method of Shi et al. [29]. Hydrogen peroxide (H_2_O_2_) content was measured following the methods described by Mittler et al. [27]. Catalase (CAT) activity and superoxide dismutase (SOD) activity were assessed following the protocols delineated by Ning et al. [25].

### 4.6. Statistical Analysis

The data underwent analysis through analysis of variance (ANOVA) employing the SPSS 10 software (SPSS Inc., Chicago, IL, USA). Varied letters are indicative of a statistically significant distinction between the transgenic line and the wild type (*p* < 0.05, one-way ANOVA).

## Figures and Tables

**Figure 1 plants-12-03166-f001:**
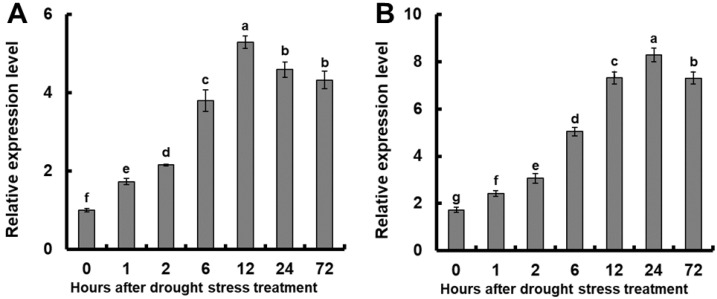
Quantitative assessment of the relative expression of *OsGSTU17* during drought stress treatment was conducted using real-time RT-PCR. Rice seedlings were subjected to an initial growth period of 7 days in the regular IRRI solution, followed by transfer to a nutrient solution containing 10% (*w*/*v*) PEG6000 for varying durations. Subsequent RNA extraction was performed on distinct segments, namely (**A**) root and (**B**) shoot tissues of the Nipponbare rice cultivar. The error bars in the figures represent the standard error (SE) derived from triplicate analyses (*n* = 3 plants). The different letters indicate a significant difference between the treatments (*p* < 0.05, one-way ANOVA).

**Figure 2 plants-12-03166-f002:**
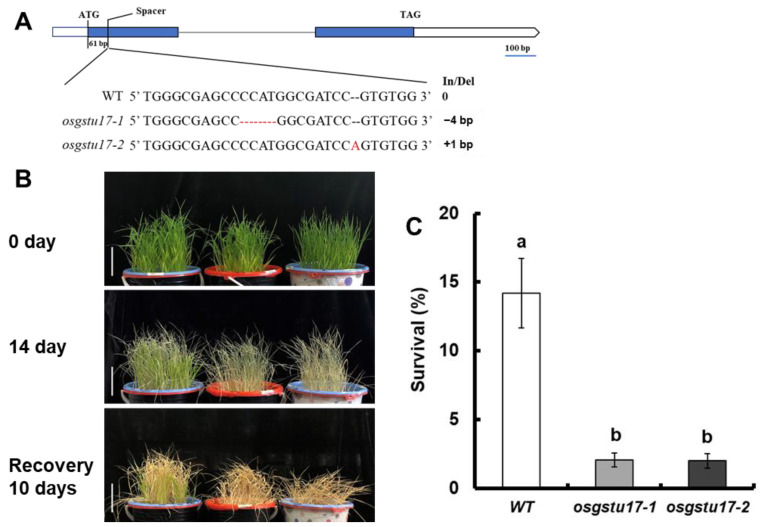
Drought stress sensitivity of *osgstu17* mutant lines at the seedling stage. (**A**) Schematic diagram of the position of the target site (spacer) by CRISPR/Cas9-induced editing is shown on the *OsGSTU17* gene structure. Sequencing outcomes of mutant alleles are aligned against the reference genome sequence, and the sizes of the insertions and/or deletions (In/Del) are indicated on the right. (**B**) Phenotype of drought-stressed plants followed by recovery. The seedlings were cultivated for a duration of 21 days under adequately irrigated conditions, employing a practice of accommodating 10 seedlings of uniform size per pot. Subsequently, irrigation was suspended for a span of 14 days, followed by a 10-day rehydration phase. A bar of 10 cm was utilized for scale reference. (**C**) Seedling survival. The count of seedlings that displayed at least one fully expanded leaf was recorded. The error bars in the figures depict the standard error (SE) derived from triplicate pots (*n* = 3 pots). The utilization of distinct letters signifies statistically significant disparities between the transgenic line and the wild type (WT) based on a significance level of *p* < 0.05, as determined by a one-way analysis of variance (ANOVA).

**Figure 3 plants-12-03166-f003:**
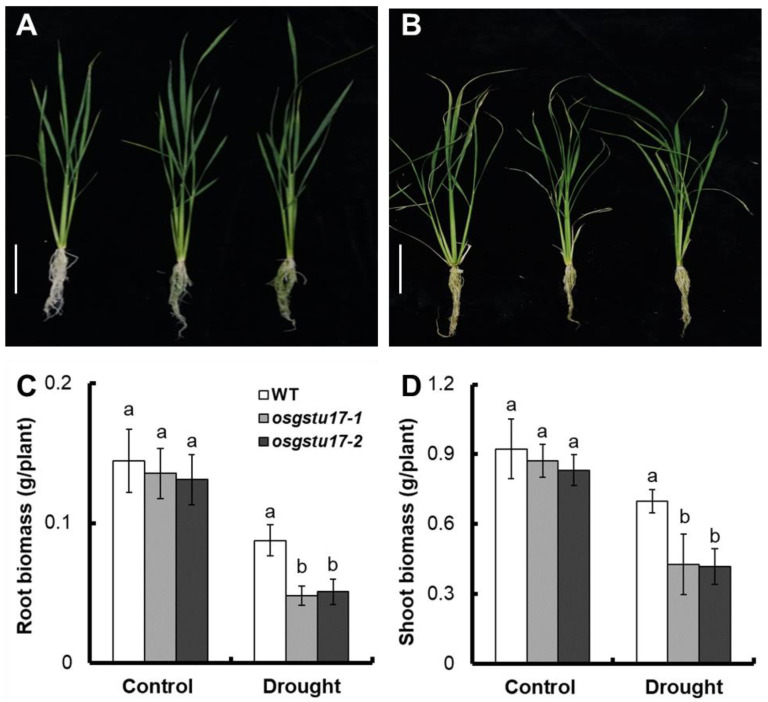
The biomass of *osgstu17* mutant lines under control and drought stress conditions. Rice seedlings were cultivated in the standard IRRI solution for a period of 2 weeks, subsequent to which they were transplanted into a nutrient solution supplemented with 10% (*w*/*v*) PEG6000 for an additional 2-week duration. Phenotype of *OsGSTU17* mutant lines grown in (**A**) control and (**B**) drought stress (10% PEG6000) conditions. Bar = 10 cm. (**C**) Root and (**D**) shoot biomass (dry weight) of plants grown in control and drought stress conditions. Error bars: SE (*n* = 4 plants). Varied letters denote a statistically significant distinction between the transgenic line and the WT as determined by a one-way analysis of variance (ANOVA) with a significance level of *p* < 0.05.

**Figure 4 plants-12-03166-f004:**
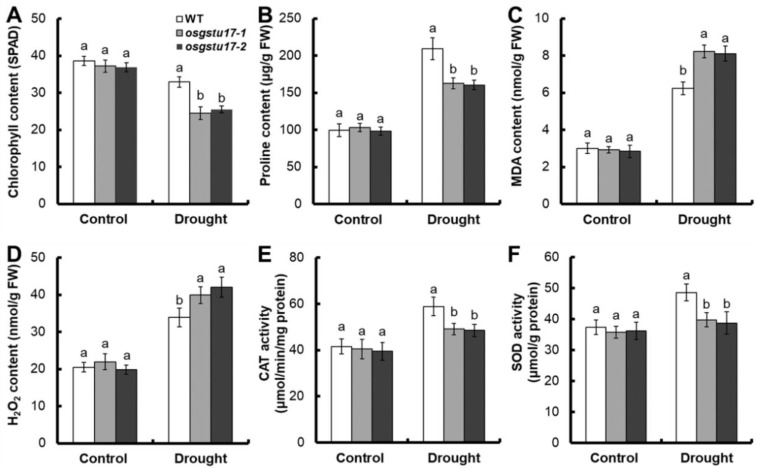
Physiological and biochemical changes in *osgstu17* mutant lines. The growth conditions and treatment protocols remained consistent with the description provided in Figure 3. (**A**) The chlorophyll content, (**B**) proline content, (**C**) MDA content, (**D**) H_2_O_2_ content, (**E**) CAT activity and (**F**) SOD activity of *osgstu17* mutant lines under control and drought stress. Error bars: SE (*n* = 4 plants). Varied letters denote a statistically significant distinction between the transgenic line and the WT as determined by a one-way analysis of variance (ANOVA) with a significance level of *p* < 0.05.

**Figure 5 plants-12-03166-f005:**
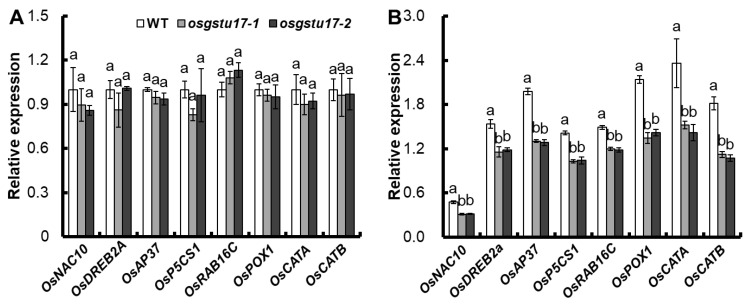
Transcript levels of stress-responsive genes in the *osgstu17* mutant lines were assessed. The growth conditions and treatment protocols remained consistent with the description outlined in Figure 3. RNA extraction was carried out from the shoot tissues of seedlings subjected to (**A**) control and (**B**) drought stress conditions. The error bars depicted in the figures represent the standard error (SE) derived from triplicate plant samples (*n* = 3 plants). Varied letters indicate statistically significant distinctions between the transgenic line and the WT, as determined by an ANOVA with a significance level of *p* < 0.05.

**Table 1 plants-12-03166-t001:** Primers used for qRT-PCR.

Gene Name	Forward Primer (5′ to 3′)	Reverse Primer (5′ to 3′)
*OsActin*	GGAACTGGTATGGTCAAGGC	AGTCTCATGGATAACCGCAG
*OsGSTU17*	TCTTCATGACGACCGGAGAG	GGTGACGATGTCAAGGTAGC
*OsNAC10*	TTCTCCTCGACGGCTCATCC	ATGGATGGCTCAGCAGATTG
*OsDREB2A*	GGCTGAGATCCGTGAACCAA	GGACCATACATTGCCCTTGC
*OsAP37*	TCCGATGTTTTGGTCCTCTG	TCCACGGTTTAGTCCATCTCATC
*OsP5CS1*	GCTGACATGGATATGGCAAAAC	GTAAGGTCTCCATTGCATTGCA
*OsRAB16C*	CCCGGCCAGCACTAAATAAG	AAACTGCACGTACATCACGACAT
*OsPOX1*	CATCCCAGCTCCCAACAA	AGACATGCCAATGGTGTGG
*OsCATA*	GCCGGATAGACAGGAGAGGT	TCTTCACATGCTTGGCTTCA
*OsCATB*	GGTGGGTTGATGCTCTCTCA	ATTCCTCCTGGCCGATCTAC

## Data Availability

All of the data generated or analyzed during this study are included in this published article.

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
