# Peer review of "OsGSTU17, a Tau Class Glutathione S-Transferase Gene, Positively Regulates Drought Stress Tolerance in Oryza sativa"

_plants, 2023, doi:10.3390/plants12173166_

Round 1
Reviewer 1 Report
Manuscript OsGSTU17, A tau class Glutathione S-transferase Gene, Positively Regulates Drought Stress Tolerance in Oryza sativa by Jinyan Li, Lijun Meng, Shuohan Ren, Chunying Jia, Ruifang Liu, Hongzhen Jiang, Jingguang Che proposes the results of regulation of Glutathione S-transferase under stress.
The manuscript is compiled according to the rules. However, it is desirable to highlight the conclusion in order to clarify the results.
Figure 1 looks a little strange. Unlike other histograms, there is no letter identification of differences.
It is also desirable to change the design of the histograms by making them in color.
To understand the differences, the correlation between gene expression and physiological rapamers would be useful, as it is difficult to understand this without appropriate processing.
In addition, the section on obtaining transgenic plants should be expanded.
It should also be clarified how many plants were used in each case, whether there were differences between them.
Author Response
Q1. The manuscript is compiled according to the rules. However, it is desirable to highlight the conclusion in order to clarify the results.
R1. Thank you. As a "Communication", we aim to articulate the finding that the disruption of OsGSTU17 diminishes rice's tolerance against drought stress, highlighting OsGSTU17's affirmative role in bolstering drought tolerance in rice. However, the precise molecular mechanisms through which OsGSTU17 engages in the regulation of drought tolerance in rice remains unclear.
Q2. Figure 1 looks a little strange. Unlike other histograms, there is no letter identification of differences.
It is also desirable to change the design of the histograms by making them in color.
R2. Thank you very much for pointing out our mistakes. We have incorporated distinct letters to denote variations between treatments. All Figures featured in our manuscript are presented in grayscale. To ensure a consistent visual style, we have abstained from utilizing coloration in the present context.
Q3. To understand the differences, the correlation between gene expression and physiological rapamers would be useful, as it is difficult to understand this without appropriate processing.
R3. Thank you for your comments and suggestions. In terms of both gene expression and physiological parameters, there is no significant distinction observed between our two transgenic lines. Drawing conclusive inferences from correlation analysis between these lines is challenging. Conversely, the levels of both malondialdehyde (MDA) and hydrogen peroxide (H2O2), known for their phytotoxicity, exhibited elevation in the osgtu17 mutant lines. In contrast, chlorophyll and proline content experienced a reduction of approximately 20%. The activities of catalase and superoxide dismutase, enzymes engaged in peroxide detoxification, also decreased by around 20 percent. Under drought conditions, a noteworthy reduction was observed in the expressions of drought stress-related genes (OsNAC10, OsDREB2A, OsAP37, OsP5CS1, OsRAB16C, OsPOX1, OsCATA, and OsCATB) within the osgtu17 mutant lines, as compared to the wild type. Moreover, under drought treatment, when compared with the wild type, alterations in the expression of physiological, biochemical, and stress-responsive genes in the transgenic lines collectively signify the involvement of OsGSTU17 in conferring rice tolerance. Serving as a "Communication," this is also the central conclusion we intend to convey.
Q4. In addition, the section on obtaining transgenic plants should be expanded. It should also be clarified how many plants were used in each case, whether there were differences between them.
R4. Thank you. We have supplemented it in Figure legends.
Reviewer 2 Report
General comment
This paper brings interesting results. The study was well conducted and the manuscript is well written.
Comments and corrections
As a suggestion, authors may benefit from checking this related paper and consider citing if appropriate (I am not an author of the paper and do not know the authors).
Xue Yang, Wu Sun, Jiang-Peng Liu, Yan-Jing Liu, Qing-Yin Zeng, Biochemical and physiological characterization of a tau class glutathione transferase from rice (Oryza sativa), Plant Physiology and Biochemistry, Volume 47, Issues 11–12, 2009, Pages 1061-1068, ISSN 0981-9428, https://doi.org/10.1016/j.plaphy.2009.07.003
Title
“OsGSTU17, A tau” -> “OsGSTU17, a tau”
Affiliations
“ Correspondence: Correspondence: “
Last sentence of the abstract is not clear; what kind of specific functions remain to be further explored?
Finally, we concluded that knocking out OsGSTU17 significantly reduces the drought tolerance of rice, OsGSTU17 could be used as a candidate gene for rice drought tolerance cultivation, but its specific function remains to be further explored.
Introduction
“By 2080, crop yield reduction caused by drought stress are expected…” -> “ By 2080, crop yield reduction caused by drought stress is expected…”
“2.1. Relative expression of OsGSTU17 under drought stress.” and “2.4. Effects of OsGSTU17 knockout on rice seedling physiological indicators.”
Delete the full stop.
Discussion
Arabidopsis needs to be in italics
References
17. Arabidopsis thaliana – italics are missing; check for other instances of missing italics in the references section
The paper was easy to read, English is fine/minor issues.
Author Response
Q1. As a suggestion, authors may benefit from checking this related paper and consider citing if appropriate (I am not an author of the paper and do not know the authors).
Xue Yang, Wu Sun, Jiang-Peng Liu, Yan-Jing Liu, Qing-Yin Zeng, Biochemical and physiological characterization of a tau class glutathione transferase from rice (Oryza sativa), Plant Physiology and Biochemistry, Volume 47, Issues 11–12, 2009, Pages 1061-1068, ISSN 0981-9428, https://doi.org/10.1016/j.plaphy.2009.07.003
R1. Thank you for your valuable suggestion. This publication is highly pertinent to our research, and we have duly cited it.
Q2. Title “OsGSTU17, A tau” -> “OsGSTU17, a tau”
R2. Thank you. Actioned.
Q3. Affiliations
“ Correspondence: Correspondence: “
R3. Accepted and actioned.
Q4. Last sentence of the abstract is not clear; what kind of specific functions remain to be further explored?
Finally, we concluded that knocking out OsGSTU17 significantly reduces the drought tolerance of rice, OsGSTU17 could be used as a candidate gene for rice drought tolerance cultivation, but its specific function remains to be further explored.
R4. Thank you. “but its specific function remains to be further explored. ” be rewritten as “However, the molecular mechanism of OsGSTU17 involved in rice drought resistance needs to be further studied.” in Abstract.
Q5. Introduction
“By 2080, crop yield reduction caused by drought stress are expected…” -> “ By 2080, crop yield reduction caused by drought stress is expected…”
R5. Thank you. Accepted and actioned.
Q6. “2.1. Relative expression of OsGSTU17 under drought stress.” and “2.4. Effects of OsGSTU17 knockout on rice seedling physiological indicators.”
Delete the full stop.
R6. Thank you. Accepted and actioned.
Q7. Discussion
Arabidopsis needs to be in italics
R7. Accepted and actioned.
Q8. References
- Arabidopsis thaliana – italics are missing; check for other instances of missing italics in the references section
R8. Accepted and actioned
Reviewer 3 Report
In this study, the authors created two mutant lines of OsGSTU17 with CRISPR and evaluated them under drought stress. The results clearly showed the two mutant lines were more secentive to drought stress and other drought-resistant genes were impacted by the silence of OsGSTU17.
A few comments?questions:
1:Why don't you create overexpression lines as well?
2. Are the mutants homozygous lines?
3: The PDF format should generated with line numbers, it is difficult for reviewer to point out where the type error or mistake is.
4: on page 7, the first paragraph of "4.1. Plasmid construction and plant transformation should be modified.
5: On page 6, Under discussion, is the "water relations" should be "water retention"?
The quality of English is very good.
Author Response
Q1. Why don't you create overexpression lines as well?
R1. Yes, knockout mutants and overexpressed lines are commonly used in plant gene function research. However, many studies only use knockout mutants or overexpressed lines (Yamaji et al., 2013; Yang et al., 2014; Chang et al., 2020). The function of a gene can be explored by knocking out or overexpressing it and its impact on plants. In this manuscript, we obtained knockout mutant lines of OsGSTU17 using the CRISPR/cas9 method.
References
Yamaji N, Sasaki A, Xia JX, Yokosho K, Ma JF. A node-based switch for preferential distribution of manganese in rice. Nat Commun. 2013, 4:2442.
Yang M, Zhang Y, Zhang L, Hu J, Zhang X, Lu K, Dong H, Wang D, Zhao FJ, Huang CF, Lian X. OsNRAMP5 contributes to manganese translocation and distribution in rice shoots. J Exp Bot. 2014, 65(17):4849-61.
Chang JD, Huang S, Yamaji N, Zhang W, Ma JF, Zhao FJ. OsNRAMP1 transporter contributes to cadmium and manganese uptake in rice. Plant Cell Environ. 2020, 43(10):2476-2491.
Q2. Are the mutants homozygous lines?
R2. Yes, as shown in Figure 2, both transgenic lines are homozygous.
Q3. The PDF format should generated with line numbers, it is difficult for reviewer to point out where the type error or mistake is.
R3. Thank you. Accepted and actioned.
Q4. on page 7, the first paragraph of "4.1. Plasmid construction and plant transformation should be modified.
R4. Thank you. Accepted and actioned.
Q5. On page 6, Under discussion, is the "water relations" should be "water retention"?
R5. Very sorry, we have deleted this sentence 'Under drought stress, regulating nitrogen supply can improve water relations and thus enhance crop adaptability'. This sentence is not very relevant to the content of this manuscript, so we have deleted it in the new manuscript. Thank you very much for your reminder. We have also checked the entire text.
Round 2
Reviewer 1 Report
The OsGSTU17 manuscript, A tau class Glutathione S-transferase Gene, Positively Regulates Drought Stress Tolerance in Oryza sativa by Jinyan Li et al., has been improved.
Answers to some comments have been fully reflected in the new version.
I think that the discussion of the mechanisms and role of this gene in the discussion was useful.
Author Response
Dear Reviewer,
Thank you very much for your careful handling of our manuscript and giving us the opportunity of a revision and resubmission. Your comments and suggestion are very valuable and highly appreciated. The followings are our responses. We hope that the revised version of the manuscript is acceptable to you.
- Reviewer 1’s Comments:
Q1. The OsGSTU17 manuscript, A tau class Glutathione S-transferase Gene, Positively Regulates Drought Stress Tolerance in Oryza sativa by Jinyan Li et al., has been improved.
Answers to some comments have been fully reflected in the new version.
R1. Thank you very much.
Q2. I think that the discussion of the mechanisms and role of this gene in the discussion was useful.
R2. Thank you for your comments and suggestions. We further discussed the possible physiological and molecular mechanisms underlying the involvement of the OsGSTU17 in rice drought resistance in section Discussion. Please refer to Discussion in the manuscript for detailed information.